# Machine Learning Heuristics on Gingivobuccal Cancer Gene Datasets Reveals Key Candidate Attributes for Prognosis

**DOI:** 10.3390/genes13122379

**Published:** 2022-12-16

**Authors:** Tanvi Singh, Girik Malik, Saloni Someshwar, Hien Thi Thu Le, Rathnagiri Polavarapu, Laxmi N. Chavali, Nidheesh Melethadathil, Vijayaraghava Seshadri Sundararajan, Jayaraman Valadi, P. B. Kavi Kishor, Prashanth Suravajhala

**Affiliations:** 1Bioclues.org, Hyderabad 500072, India; 2Khoury College of Computer Sciences, Northeastern University, Boston, MA 02115, USA; 3Molecular Signaling Lab, Faculty of Medicine & Health Technology, Tampere University, 33100 Tampere, Finland; 4Amity Institute of Microbial Technology, Amity University, SP-1 Kant Kalwar, NH11C, RIICO Industrial Area, Rajasthan 303002, India; 5Amrita School of Biotechnology, Amrita Vishwa Vidyapeetham, Clappana 690525, India; 6Department of Computer Science, FLAME University, Pune 412115, India; 7MNR Foundation for Research & Innovation, MNR Medical College and Hospital, Fasalwadi, Sangareddy, Hyderabad 502294, India

**Keywords:** oral cancer, machine learning, gene prioritization, genomic datasets, data mining

## Abstract

Delayed cancer detection is one of the common causes of poor prognosis in the case of many cancers, including cancers of the oral cavity. Despite the improvement and development of new and efficient gene therapy treatments, very little has been carried out to algorithmically assess the impedance of these carcinomas. In this work, from attributes or NCBI’s oral cancer datasets, viz. (i) name, (ii) gene(s), (iii) protein change, (iv) condition(s), clinical significance (last reviewed). We sought to train the number of instances emerging from them. Further, we attempt to annotate viable attributes in oral cancer gene datasets for the identification of gingivobuccal cancer (GBC). We further apply supervised and unsupervised machine learning methods to the gene datasets, revealing key candidate attributes for GBC prognosis. Our work highlights the importance of automated identification of key genes responsible for GBC that could perhaps be easily replicated in other forms of oral cancer detection.

## 1. Introduction

Oral cavity cancer (OCC) is the tenth most common malignant tumor in the world and the third most common in southeast Asia. The common subsite recorded in OCC in third world countries, especially in Indian communities, is gingivobuccal cancer (GBC) constituting about 40% of all cases, whereas the cases diagnosed in the western world are about 10% [1]. They are usually associated with delayed clinical detection, poor prognosis, absence of specific biomarkers for the disease, and expensive therapeutic alternatives [2]. The GBC comprises buccal mucosa, gingivobuccal sulcus, alveolus, and retromolar area cancers and is commonly seen in younger patients. While certain precancerous conditions and lesions such as submucous fibrosis, leukoplakia, and erythroplakia are known causes, dietary deficiencies such as iron, Vitamins A, C, and E are associated with oral cancers. The processes such as segregation of chromosomes, genomic copy number, loss of heterozygosity, telomere stabilities, regulations of cell-cycle checkpoints, DNA damage repairs, and defects in Notch signaling pathways are involved in causing oral cancer [3]. Malignant odontogenic tumors emanate de novo within jawbones, from epithelium contained within cyst linings, or from the malignant transformation of benign odontogenic tumors. The lesions most commonly are the primary intraosseous carcinomas, including the mucoepidermoid carcinoma arising within the bone and the ameloblastic carcinoma [4]. The WHO classification of odontogenic carcinoma dissects malignant ameloblastoma from primary intraosseous carcinoma [5]. As diagnosis is entrenched by a biopsy of the jaw lesion, the definitive analysis prospective is of a usually poor outcome. Early signs and symptoms include soreness or pain in jaws, which could extend through chewing/swallowing followed by loosening teeth and bleeding from the mouth. While a good examination is heralded by visualization in the buccal mucosa, the current high-end transoral robotic surgeries [TORS], besides vaccines, have been in use [6].

Over the last decade, several treatments have been utilized with the consistent use of effective gene therapies. Discoveries about how changes in the DNA of cells in the oral cavity and oropharynx cause these cells to become cancerous are being applied to experimental treatments intended to reverse these changes. For example, clinical trials are testing whether it is possible to replace abnormal tumor suppressor genes (such as the p53 gene) of oral cancer cells with a normal copy to restore normal growth control [7]. Machine learning is a computational method that improves performance to make accurate predictions when data analysis and statistical methods do not have enough information about the underlying distribution of data [8]. Furthermore, from our previous experience, machine learning algorithms have been applied to various fields in genomics [9], healthcare [10], computer vision [11], etc. As the applications of these methods have assisted the precision medicine scale, this would eventually bridge the gaps in oral squamous cell carcinoma [12]. Ahmed et al. [13] investigated these methods from the Artificial Intelligence (AI) dental imaging perspective. The metadata constituting characteristics, study and control groups were extracted for feature selection paradigms, which resulted in understanding the implications of the OSCC. Nevertheless, AI could predict failures to assess the clinical performance in such carcinomas [13]. Through the use of statistical methods, the variables (weights) in the algorithm undergo systematic updates representing the distribution of the training data during the training phase. The test phase presents unique, unseen data to the same algorithm weights and makes a classification/prediction for this new data point. As these algorithms can help uncover key insights within data mining projects, subsequent decision-making drives can ideally impact key growth metrics.

Oral cancer prognosis is one of the burgeoning problems, and our work employing machine learning heuristics could lay emphasis on piloting candidate biomarkers. As diagnosis could be better aided for prognosis and theranostics, survival and therapies must be in place, and despite strategic improvements in these areas, this is still in infancy. Various phenotypes associated with oral cancer, such as oral squamous cell carcinoma (OSCC) and early-stage diagnosis, are still in the realm of early-stage detection. Alabi et al. [12] have reviewed the challenges and examined the need for deep-learning heuristics for the proper management of oral cancer. In the present work, we aimed to identify a key candidate signature for oral cancer/gingivobuccal phenotype from various datasets that are mined from NCBI, wherein these could be considered as prognostic signature biomarkers. We employed a mixture of supervised and unsupervised algorithms and attempted to understand key attributes for prognosis of oral cancer. While supervised methods are much simpler and straightforward to use for our study, we wanted to briefly touch upon the usefulness of unsupervised methods for motivating further research with this combination of data. A detailed gist of results employing a Support Vector Machine (SVM), Naïve Bayes, Decision trees, Multi-Layer Perceptron, Logistic Regression, and K Means (unsupervised) are discussed.

## 2. Materials and Methods

### 2.1. Datasets and Transformation

We used datasets for four genes related to oral cancer: *PIK3CA, KRAS, TP53* and Gingival. The dataset from NCBI (www.ncbi.nlm.nih.gov, last accessed on 27 October 2022) searches was screened with the following five features: (i) name, (ii) gene(s), (iii) protein change, (iv) condition(s), clinical significance (last reviewed). TP53 and Gingival have an additional Review Status feature. The number of samples varies for each dataset: PIK3CA has 544 instances, KRAS has 330 instances, TP53 has 2186 instances, and Gingival has 2107 instances (Appendix A; Figure 1).

We transformed alphanumeric features into categorical features for the application of the following Machine Learning Algorithms (as given below in the section Classifier Design and Training). The first instance of data values of protein change, condition(s), clinical significance (last reviewed), and review status was used. Then, the data values of features such as gene(s), protein change, condition(s), clinical significance (last reviewed), and review status were converted into numeric keys using Preprocessing and Transformation classes in Scikit-learn. Binary and numeric weightages were assigned to each feature, including protein change, condition(s), clinical significance (last reviewed), and review status to evaluate the performance based on data annotations.

### 2.2. Experiments

We performed four experiments for PIK3CA and KRAS datasets and six experiments for TP53 and Gingival using different combinations of features. The following six experiments separately used one of the following four features: (i) all the features in a dataset, (ii) only binary features, (iii) only non-binary features, (iv) all features except review status (for datasets (TP53, Gingival) that contains review status as a feature), (v) only non-binary features with no review status (for datasets (TP53, Gingival) that contains review status as a feature), and (vi) only binary features with no review status (for datasets (TP53, Gingival) that contains review status as a feature).

### 2.3. Classifier Design and Training

We used six major classifiers to train and test the model: (i) Support Vector Machine, (ii) Naïve Bayes, (iii) Decision trees, (iv) Perceptron, (v) Logistic Regression, and (vi) K Means (unsupervised). We randomly split the dataset to use 80% for training and 20% for testing. We used off-the-shelf algorithms implemented in Scikit-learn for these experiments and used other libraries, such as NumPy, Pandas, and Matplotlib available in Python 3.10.7. While unsupervised algorithms are hard to implement on such data, we used only K Means for a flavor of unsupervised learning. Further analyses with algorithms, such as K Medoids, PCA, etc., are left for future work.

### 2.4. Performance Evaluation

Evaluating the performance of learning algorithms is a central aspect of machine learning. We used an 80-20 train-test split to test the performance of the predictive and classification models. To mitigate the overfitting problem, the following measures were used to evaluate the performance of six classifiers based on accuracy, which is defined as the percentage of correct predictions for the test data. It can be calculated by dividing the number of correct predictions by the number of total predictions. The measure is defined as follows:*Accuracy*  =  [*TP*  +  *TN*]/[*TP*  +  *FN*  +  *FP*  +  *TN*]
where TP—True Positives (positive samples classified correctly as positive), TN—True Negatives (negative samples classified correctly as negative), FP—False Positives (negative samples predicted wrongly as positive), and FN—False Negatives (positive samples predicted wrongly as negative). The precision and recall were achieved with inherent accuracy.

## 3. Results and Discussion

### 3.1. PIK3CA among the Select Genes with Highest Accuracy

One of the interesting findings we attempted in our study was to identify gene datasets that are significantly enriched from machine learning heuristics. We observe that there is a significant amount of attribute fitting with instances taken up from all datasets. While all instances were used and compared across all algorithms to further gain insight into this, the accuracies were tabulated accordingly (Figure 2; Appendix A). For PIK3CA, experiment (i) accuracy varies between 78% (decision tree) and 48% (Naïve Bayes). For experiment (ii), accuracy varies between 67% (MLP) and 41% (Naïve Bayes). For experiment (iii), accuracy varies between 77% (decision tree) and 44% (Naïve Bayes). On the other hand, for KRAS, experiment (i) accuracy varies between 62% (decision tree) and 27% (K Means). For experiment (ii), accuracy varies between 62% (decision tree) and 17% (Naïve Bayes). For experiment (iii), accuracy varies between 53% (decision tree) and 18% (Naïve Bayes). Whereas TP53 showed variable changes, for experiment (i) accuracy varies between 61% (MLP) and 35% (K Means). For experiment (ii), accuracy varies between 56% (SVM, MLP and decision tree) and 35% (K Means). For experiment (iii), accuracy varies between 55% (MLP) and 8% (Naïve Bayes). For experiment (iv), accuracy varies between 57% (MLP) and 34% (K Means). Additionally, for experiment (v), accuracy varies between 50% (decision tree) and 21% (K Means). For experiment (vi), accuracy varies between 51% (decision tree, logistic regression, MLP, SVM) and 35% (K Means). For gingival datasets, experiment (i), accuracy varies between 63% (MLP) and 29% (K Means); for experiment (ii), accuracy varies between 49% (MLP) and 29% (K Means); for experiment (iii), accuracy varies between 63% (decision tree) and 29% (K Means); for experiment (iv), accuracy varies between 54% (MLP) and 29% (K Means); for experiment (v), accuracy varies between 52% (MLP) and 29% (K Means); and for experiment (vi), accuracy varies between 40% (decision tree, logistic regression, MLP, SVM) and 30% (K Means clustering). From the above results, it is evident that only experiment (i) is shown to have the highest accuracy when compared with other experiments from (ii) to (vi) (Table 1). We present accuracies with 5-fold cross-validation using different algorithms on the 4 genes in Table 2.

What we aimed to achieve from our pilot study is to employ gene selection and ask whether or not the lesser-known changes in attributes can, by choice, be ignored for further prognosis. In other words, in nature, are there any genes that are repetitively expressed with inherent changes attempted in our machine learning heuristics [14]. The virtual experiments on ML heuristics that we employed set a base for oral cancer prognosis. However, there is a dearth of well-annotated or informative attributes, which is a major limitation of our work. Theoretically, with more instances and genes segregated from the attributes, we could have received a better performance and overcome the overfitting problem, albeit the fact that our finding of the relevant four genes augments the hypothesis that it may not always be true. Our experiments and framework can further be extended to reveal the effects of key attributes from genetic data and be applied to predict outcomes, such as the chances of survival, recurrence, etc. On the other hand, some work has been seen around survival risk stratification [15] and survival prediction [16] using similar machine learning-based methods. The majority of these works have patient datasets collected for several years. Even as these yield bona fide results, they could be prone to biases. We found the application of Principal Component Analysis (PCA) and other techniques for data reduction to be prevalent in multiple studies.

Initially, we ran the experiments with the same data splits and the same machine learning algorithms using the java-based package Weka [17,18]. While we found the results to be overfitting to our data, we speculate that Weka assigns every non-numeric instance to be a unique key and processes them individually. For example, when A-B is arranged as B-A in the dataset (without ordering-sensitive features), Weka is unable to break them and considers them as two keys instead of one. A clear limitation for this approach is indicative of certain data types, as it also relies heavily on data annotation. Having data annotated (manual and program) to account for such orderings, we find that our models do not overfit and perform better, which could be the plausible reason why many annotated cancer datasets have. This also agrees that the scarcity of publicly available image datasets may impede early patho-significant diagnoses for cancers taking the machine learning paradigm [19]. Although Kaggle has some datasets (https://www.kaggle.com/datasets/shivam17299/oral-cancer-lips-and-tongue-imageslast, accessed on 30 October 2022), the size is limited and might be underfitting in the present context. On the other hand, to overcome the overfitting and failed model as we postulated, deep-learning models could bring great promise for an accurate prognosis if in case the datasets have tumorigenic data, infiltrating lymphocytes and multiclass labeling, which can herald predicting disease states [13]. Such data could then be divided into risk groups and then differentiate the data from a good to a poor prognosis. Having said this, deep-learning clubbed with precise detection may then be used to identify oral cancer datasets, albeit the fact that there must be high-end computability to identify multidimensional datasets.

### 3.2. Scatter Plot for K-Means

We present a scatter plot for PIK3CA using the features ProteinChange_keys and Binary Scoring CS. We show five classes represented by colors red, blue, green, yellow, cyan, while centroids are represented by ‘X’. The classes represented by red, yellow, blue, and cyan form tight clusters around centroids, showing that the clusters capture the underlying data distribution well, while the cluster represented by green is slightly far from the calculated centroid (Figure 3). Additional scatter plots are provided in the Appendix A.

Given the success of multimodal algorithms [20], we believe our analysis can be further strengthened by using microscopic images of cells from the buccal cavity alongside annotated genetic data. Using electron microscopy and image segmentation algorithms, it is now possible to segment the image up to the cellular level, precisely pinpointing the areas of carcinoma. Such precise positions can help prevent the pitfalls of annotation errors, making our analysis more robust. We speculate that such analysis can also aid in predicting the early onset of cancers [21,22]. Taken together, our analyses could provide early roads for prognosis where these genes could aid as key candidates. As diagnosis could be better aided for prognosis and theranostics, survival and therapies must be in place, and despite strategic improvements in these areas, this is still in infancy. Machine learning and artificial intelligence (AI) aided methods have enhanced early detection in reducing mortality and morbidity. Indefatigably, there are not many metadata-based machine learning heuristics assessing the impedance of these carcinomas. In summary, we presented a machine learning-based approach to predict the gene dataset, revealing key candidate attributes for GBC prognosis.

## 4. Conclusions

Machine learning and Artificial Intelligence (AI) aided methods have enhanced early detection in reducing mortality and morbidity. Indefatigably, there are not many metadata-based machine learning heuristics assessing the impedance of these oral carcinomas. In summary, we presented a machine learning-based approach to predict the gene dataset, which reveals key candidate attributes for GBC prognosis. We have attempted to fill these gaps by performing and labeling classes, and accurate identification of viable attributes for such cancers. Furthermore, we found that deterministic methods perform well with limited data. In contrast, non-deterministic methods excel in performance with large datasets, wherein supervised learning methods perform better than unsupervised methods. Nonetheless, our experiments had more supervised methods than unsupervised ones, which we wanted to establish the use case for such an analysis. We argue that a multitude of unsupervised and semi-supervised methods might be able to better model these data distributions that seldom have accurate annotations. However, this may be due to the lack of machine learning heuristics which could be used as models and vice versa for a better-modeled framework.

## Figures and Tables

**Figure 1 genes-13-02379-f001:**
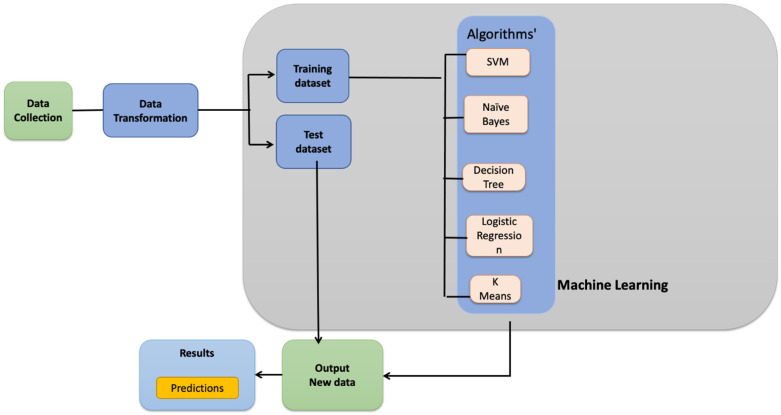
Machine Learning pipeline used for our analysis.

**Figure 2 genes-13-02379-f002:**
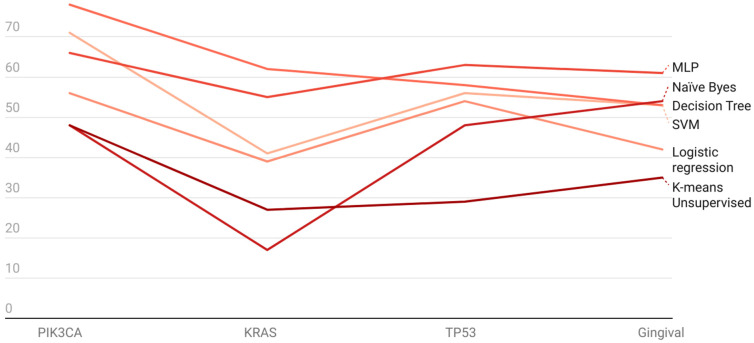
Machine Learning accuracies for vivid datasets exploited using various experiments.

**Figure 3 genes-13-02379-f003:**
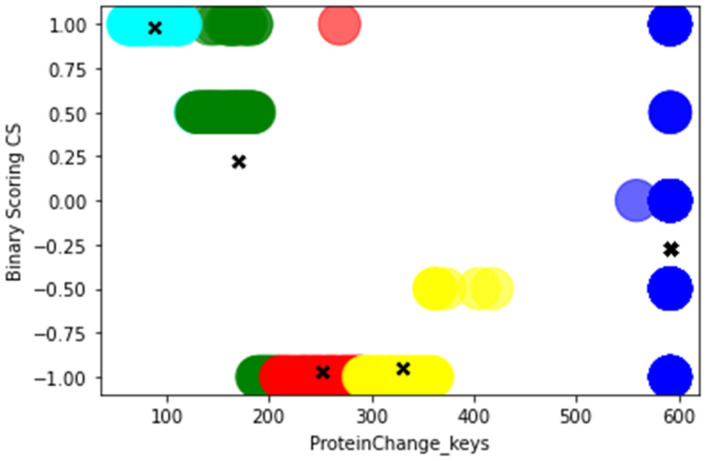
Scatter plot between ProteinChange_keys and Binary Scoring CS of gene PIK3CA with K-Means.

**Table 1 genes-13-02379-t001:** ML Accuracies for each Oral Cancer Gene.

		ML Algorithms [Accuracies %]					
		SVM	MLP	Logistic Regression	Naïve Byes	Decision Tree	K-Means Unsupervised
Genes	PIK3CA	71	66	56	48	78	48
	*KRAS*	41	55	39	17	62	27
	*TP53*	56	63	54	48	58	29
	Gingival	53	61	42	54	53	35

**Table 2 genes-13-02379-t002:** Accuracies through 5-fold classification on different algorithms for the 4 genes.

Genes	SVM	MLP	Logistic Regression	Naïve Bayes	Decision Tree	K-Mean
*PIK3CA*	88%	83%	76%	100%	87%	21%
*KRAS*	82%	70%	74%	19%	92%	8%
*TP53*	51%	57%	63%	89%	57%	53%
Gingival	47%	48%	53%	95%	45%	55%

## Data Availability

All data is available in the form of Supplementary Information.

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
