# Peer review of "Machine Learning Heuristics on Gingivobuccal Cancer Gene Datasets Reveals Key Candidate Attributes for Prognosis"

_genes, 2022, doi:10.3390/genes13122379_

Round 1

Reviewer 1 Report

The article has some merits to the literature; however, here are my comments to improve the manuscript.

Abstract

1.      Please mention the aim of the study

2.      Please mention some detail about samples (like different machine learning used and numbers)

3.      Results need to be mention

4.      The conclusion concerning the aim of the study needs to be mentioned

Introduction

5.      More background literature needs to add

6.      The aim of the study should be mentioned in a single line (Considering all machine learning used).

7.      Line 64 Elaborate, complete form AI(as it appears first in the document)

Methods

8.      Please mention where and how the data sets are obtained

Results and Discussion

9.      The discussion should be compared with literature to the current finding and should give explanations for the results.

10.  Please mention and compare with some previous literature

11.  Please discuss how this method is an advantage in detecting cancer from clinical images (As mentioned in lines 63-64)

12.  Please consult the benefits over clinical diagnosis

Conclusion

13.  The conclusion paragraph part needs to be rewritten; the conclusion should be based on the aim of the study

14.  Most of the parts mentioned in the conclusions can be moved to the introduction or discussion

Author Response

The article has some merits to the literature; however, here are my comments to improve the manuscript.

Abstract

  1.     Please mention the aim of the study
  2.     Please mention some detail about samples (like different machine learning used and numbers)
  3.     Results need to be mention
  4.     The conclusion concerning the aim of the study needs to be mentioned

Thank you very much for your useful suggestion. The Abstract is revisited with suggested changes.   

Introduction

  1.     More background literature needs to add
  2.     The aim of the study should be mentioned in a single line (Considering all machine learning used).
  3.     Line 64 Elaborate, complete form AI(as it appears first in the document)

Thank you, we have discussed these changes and highlighted the text.  The aim is also reinstated in one line and it reads as follows:

Various phenotypes associated with oral cancer such as oral squamous cell carcinoma (OSCC) and early stage diagnosis  are still in the realm of early-stage detection.  Alabi et al. (12) have reviewed the challenges and examined the need for  deep learning heuristics for proper management of oral cancer.   In the present work, we aimed to identify key candidate signature for oral cancer/gingivobuccal phenotype from various datasets that are mined from NCBI, wherein these could be considered as prognostic signature biomarkers we employed a mixture of supervised and unsupervised algorithms and attempted to understand key attributes for prognosis of oral cancer. While supervised methods are much simpler and straightforward to use for our study, we wanted to briefly touch upon the usefulness of unsupervised methods for motivating further research with this combination of data

Methods

  1.     Please mention where and how the data sets are obtained

Thank you, it reads as follows: We used datasets for four genes related to oral cancer: PIK3CA, KRAS, TP53 and Gingival. The dataset from NCBI ( www.ncbi.nlm.nih.gov last accessed on October 27, 2022) searches were screened has with  the following five features: (i) name, (ii) gene(s), (iii) protein change, (iv) condition(s), clinical significance (Last reviewed). TP53 and Gingival have an additional Review Status feature. Number of samples vary for each dataset: PIK3CA has 544 instances, KRAS has 330 instances, TP53 has 2186 instances and Gingival has 2107 instances (Table S1).

Results and Discussion

  1.     The discussion should be compared with literature to the current finding and should give explanations for the results.
  2. Please mention and compare with some previous literature

Thank you, we have cited reference of 21 and compared this multi-modal analysis as we hardly could find such results associated with it. We have subtly remodified the sentence. Hope this is unto your expectations. 

  1. Please discuss how this method is an advantage in detecting cancer from clinical images (As mentioned in lines 63-64)

Thank you. We have added the following sentence

Although Kaggle has some datasets ( https://www.kaggle.com/datasets/shivam17299/oral-cancer-lips-and-tongue-imageslast accessed on October 30, 2022), the size is limited and might be underfitting in the present context. 

  1. Please consult the benefits over clinical diagnosis

We have further added these points

Taken together, our analyses could provide early in roads for prognosis where these genes could aid as key candidates. As diagnosis could be better aided for prognosis and theranostics, survival and therapies must be in place and despite strategic improvements in these areas, this is still in infancy. Machine learning and artificial intelligence (AI) aided methods have enhanced early detection in reducing  mortality and morbidity. Indefatigably, there are not many metadata based machine learning heuristics assessing the impedance of these carcinomas. In summary, we presented a machine learning based approach to predict the gene dataset which reveals key candidate attributes for GBC prognosis.

Conclusion

  1. The conclusion paragraph part needs to be rewritten; the conclusion should be based on the aim of the study
  2. Most of the parts mentioned in the conclusions can be moved to the introduction or discussion

Thank you for your useful suggestions, we have paraphrased it. 

*******************

Reviewer 2 Report

Authors employed a mixture of supervised and unsupervised algorithms and attempted to understand key attributes for prognosis of oral cancer. Nevertheless, the proposed methods are old and state-of-the-art methods are not used. The work also lacks novelty. To this end, the authors must consider the following improvements to be included in the manuscript.

1. Use supervised and unsupervised learning methods proposed in the past two years.

2. Compare the machine learniong approaches utilized against deep learning approaches.

3. Clearly describe the novelty of the work and the specific contribution of the work.

4. Evaluate the methods with more statistical metrics, not only accuracy. Addidtionally, cross validation must be used for each metrics.

5. Provide the output of the unsupervised learning approach interms of figure.

Author Response

Responses to Reviewer 2

 Comments and Suggestions for Authors

Authors employed a mixture of supervised and unsupervised algorithms and attempted to understand key attributes for prognosis of oral cancer. Nevertheless, the proposed methods are old and state-of-the-art methods are not used. The work also lacks novelty. To this end, the authors must consider the following improvements to be included in the manuscript.

  1. Use supervised and unsupervised learning methods proposed in the past two years.

Thank you for your kind suggestions. We have indeed used both approaches but provided an indefatigable extension of why such datasets fail with vivid datasets such as these. 

  1. Compare the machine learning approaches utilized against deep learning approaches.

In our humble opinion, deep learning with discrete datasets cannot be employed as we have a limited number of definitive variables and the hidden variables are limited for learning. We explained this in the penultimate section. However, we have paraphrased sections based on reviewer 1’s comments who agreed to this. We have also made subtle changes as per your suggestions.  Hope it is to your expectations. A supplementary information is also rendered. 

  1. Clearly describe the novelty of the work and the specific contribution of the work.

The novelty of our work is, given a small subset of discrete variables and small compendium of datasets, can a machine predict the best instance of key candidate signature for oral cancer?  This is merely based on sequence/gene datasets NOT image datasets which otherwise CNN could have been employed. 

  1. Evaluate the methods with more statistical metrics, not only accuracy. Additionally, cross validation must be used for each metrics.

Thank you for your response. We have clarified/included a 5-fold CV for our results.

  1. Provide the output of the unsupervised learning approach in terms of figure.

Thank you. We have included a few samples in the main text and supplementary information. 

Round 2

Reviewer 1 Report

The manuscript is now improved, all the best to the authors.

Reviewer 2 Report

Authors have addressed my questions.